# Efficacy of an ultrasound training program for nurse midwives to assess high-risk conditions at labor triage in rural Uganda

Sachita Shah[1]*, Nicole Santos[2], Rose Kisa[3], Odida Mike Maxwell[3], Jude Mulowooza[3], Dilys Walker[2,4], Krithika Meera Muruganandan[5]

1 Department of Emergency Medicine, University of Washington School of Medicine, Seattle, WA, United States of America, 2 Institute for Global Health Sciences, University of California San Francisco, San Francisco, California, United States of America, 3 Iganga General Hospital, Iganga, Uganda, 4 Department of Obstetrics, Gynecology & Reproductive Sciences, University of California San Francisco, San Francisco, California, United States of America, 5 Department of Emergency Medicine, Boston Medical Center, Boston University School of Medicine, Boston, Massachusetts, United States of America

* Sachita.shah@gmail.com

**Data Availability Statement:** Data underlying the study are available on figshare (https://doi.org/10.6084/m9.figshare.12494642).

## Abstract

Many high-risk conditions of pregnancy are undetected until the time of delivery in low-income countries. We developed a point-of-care ultrasound training protocol for providers in rural Uganda to detect fetal distress or demise, malpresentation, multiple gestation, placenta previa, oligohydramnios and preterm delivery. This was a mixed-methods study to evaluate the 2-week training curriculum and trainees' ability to perform a standard scanning protocol and interpret ultrasound images. Surveys to assess provider confidence were administered pre-training, immediately after, and at 3-month follow up. Following lecture and practical demonstrations, each trainee conducted 25 proctored scans and were required to pass an observed structured clinical exam (OSCE). All images produced 8 weeks post course underwent blinded review by two ultrasound experts to assess image quality and to identify common errors. Key informant interviews further assessed perceptions of the training program and utility of point-of-care ultrasound. All interviews were audio recorded, transcribed, and reviewed by multiple readers using a content analysis approach. Twenty-three nurse/nurse midwives and two physicians from one district hospital and three health centers participated in the training curriculum. Confidence levels increased from an average of 1 point pre-course to over 6 points post-course for all measures (maximum of 7 points). Of 25 participants, 22 passed the OSCE on the first attempt (average score 89.4%). Image quality improved over time; the final error rate at week 8 was less than 5%, with an overall kappa of 0.8–1 for all measures between the two reviewers. Among the 12 key informant interviews conducted, key themes included a desire for more hands-on training and longer duration of training and challenges in balancing clinical duties with ability to attend training sessions. This study demonstrates that providers without previous ultrasound experience can detect high-risk conditions during labor with a high rate of quality and accuracy after training.

**Funding:** DW as principal investigator for the PTBi study (parent study for this one)received funding for the study from the Bill and Melinda Gates Foundation and shared funding with co-investigators under subcontract for authors SS/NS/JM/RK. https://www.gatesfoundation.org/ The sponsors played no role in study design, data collection or analysis, decision to publish or preparation of the manuscript.

**Competing interests:** No authors have competing interests.

**Abbreviations:** BPD, Biparietal Diameter; HC, Head Circumference; FL, Femur Length; TCD, Trans Cerebellar Diameter; EGA, Estimated Gestational Age; LMIC, Low- and Middle-Income Countries; POCUS, Point-of-Care Ultrasound; HC, Health Center; DH, District Hospital; OSCE, Observed Structured Clinical (Ultrasound) Exam; KII, Key Informant Interview; TOT, Training of Trainers; QA, Quality Assurance.

# Introduction

In low- and middle-income countries (LMIC), national standards of obstetric care do not routinely include antenatal ultrasound exams to identify high-risk conditions. Despite recommendations by the World Health Organization (WHO) for a minimum of eight antenatal visits per pregnancy [1], only 36% achieve this metric in LMICs. For many women around the world, the first contact with care may be during labor [2–4]. Physical exam for certain high-risk conditions such as fetal distress, oligohydramnios, placenta previa, and multiple gestation, as well as the determination of preterm labor provides limited certainty. Much of the perinatal care provided in LMICs is by non-physicians, further complicating the ability to make these time-critical diagnoses [5–7]. Early identification, even during labor, of high-risk conditions of pregnancy may allow for better maternal and neonatal outcomes [1, 8]; however, lack of complete antenatal care and limited access to ultrasound underscore the need to develop other strategies to identify and manage these complications, including training midwives and others to perform obstetric ultrasound.

Improved portability, durability and affordability of ultrasound technology may make ultrasound-assisted diagnosis a prime candidate to address the limitations of the physical exam [8]. Ultrasound is widely considered the mainstay of diagnostic imaging in pregnancy, and is the preferred method of diagnosis for multiple gestation, oligohydramnios and placenta previa [9, 10]. Given widespread "task-sharing" of nurse midwives as primary providers in antenatal clinic, they are an ideal population in which to study benefit of ultrasound to diagnose high-risk conditions of pregnancy [11].

No prior studies have examined the aptitude of nurse midwives, who conduct the majority of obstetric care in resource-limited environments, to perform point-of care ultrasound (POCUS) at labor triage to identify high-risk conditions just before birth. Previous studies have demonstrated the utility of POCUS training in other LMIC contexts: clinical officers can accurately use ultrasound to evaluate lower respiratory infections after a 12-hour training [12]; emergency nurse practitioners can successfully diagnose a variety of emergency conditions using POCUS [13]. One prior study of nurse midwives in Zambia showed ability to scan pregnant patients at various gestational ages (not at labor triage) for basic information, however with poor ability to perform any gestational age measures [14]. Moreover, a cluster-randomized trial of antenatal ultrasound performed in a clinic setting in the second or early third trimester in resource-limited countries demonstrated no change in mortality outcome for mothers or stillbirth rate, and no change in referral patterns from ANC clinic visits for hospital delivery for complicated pregnancies [15].

This study evaluated a novel curriculum for novice ultrasound users in Eastern Uganda to diagnosis high-risk conditions at the point of labor triage using POCUS. We assessed the accuracy of images, learner confidence and changes in clinical ultrasound skills after the training intervention. We used qualitative methods to identify best practices for training in this setting.

# Materials and methods

## Study design and setting

This ultrasound training was conducted as part of a larger study (termed "parent study" hereafter) assessing the impact of POCUS on identification of 6 obstetric complications (preterm birth, oligohydramnios, placenta previa, multiple gestation, malpresentation, fetal distress or stillbirth) at one public district hospital (DH) and three health centers (HC) in Busoga Region, Eastern Uganda. The objective at the DH focused on accurate assessment of the conditions of interest, while the HC-level component assessed changes in referral. In both study arms, data

were collected on standardized study tools, including a clinical assessment/ultrasound check-list for each woman assessed.

Based on maternity register data from March 2016 to March 2017, the DH had a delivery volume of approximately 664 births/month, a caesarean section rate of 27% and an estimated 20% of deliveries had one or more of the 6 high risk conditions featured in this training. These data were collected as part of a cluster randomized controlled trial aimed at improving intra-partum and immediate newborn quality of care [16]. Three regional HCs were also included as study sites, with lower delivery volumes (approximately 70 deliveries/month at each HC) and no Cesarean capacity at the time of study initiation. The 6 obstetric conditions of interest warrant referral from the HC to the DH for management unless delivery is imminent. Ultra-sound at labor triage was not present at any of the study sites prior to the start of this study, and was only available in the outpatient department (separate building from maternity/labor triage) of the DH.

## Participants

Invited participants for the training were selected if they provide assessment, diagnostic, and management services to women who present to the maternity ward for care at one of the four facilities. The academic qualification of staff cadres recruited for the training were diploma or certificate holders (midwives or registered nurses) or higher. Trainees led participant enroll-ment and clinical assessment for the parent study; therefore, we aimed to train at least 3 mid-wives at each of the 3 HCs and 9 midwives at the DH to ensure that each shift could be covered by at least one study-trained midwife. All participants provided informed written consent prior to training implementation and qualitative interviews.

We trained a total of 23 nurse midwives and 2 physicians. The DH trainees included 2 physicians and 9 nurse midwives. The 2 physicians were included in training so that they were familiar with the parent study protocol; however, they were not involved with partici-pant recruitment and daily scanning since the intervention was designed to be midwife-led. The HC trainees had no physicians, 11 nurse midwives, 2 nurses who attended births but were not certified in midwifery, and 1 nurse who was a research study nurse performing the ultrasound portions of the study. Demographic information on participants is shown in Table 1.

**Table 1. Participant demographics.**

| N = 25 | % (n) |
|---|---|
| Provider type | |
| Nurse midwives | 80% (20) |
| Nurses | 12% (3) |
| Medical Doctors | 8% (2) |
| Gender | |
| Female | 92% (23) |
| Male | 8% (2) |
| Years Post Schooling avg (sd) | |
| Nurse midwives | 8.7 (5.1) |
| Nurses | 3 (1.7) |
| Medical Doctors | 8.5 (4.9) |

## Intervention description and roll-out

Training was rolled out in two phases in which Phase 1 clinicians from the DH (n = 11) underwent a 2-week training including both didactic and practical components. A second Phase 2 cohort of clinicians from the HCs (n = 14) completed their training 3 months later. The two training periods were October 2018 for the DH and January 2019 for the three HCs. This training timeline was based on sample size targets of the parent study.

Three DH nurse midwives from Phase 1 who expressed enthusiasm for teaching and had excellent ultrasound skills were chosen by the research team to undergo a 1-day "training of trainers (TOT)" just prior to the second cohort of training. These three Master Trainers served as resources for the Phase 2 trainees, with one Master Trainer being assigned to one HC.

The two trainings were delivered with some key differences based on lessons learned in the first training. The initial DH Phase 1 training period consisted of daily hour-long lectures in a large group of all trainees, daily hands-on scanning practice in antenatal clinic and with pregnant volunteers, and subsequent mock enrollments in labor triage to attain 25 practice scans for each of 11 course participants. In some cases, participants continued to provide clinical care while pulling away to practice scanning in this first cohort.

The Phase 2 training was modified based on lessons learned from Phase 1 and contextual differences between the study sites. Phase 2 participants traveled away from clinical duties at their respective HCs to undergo an intensive 2-day training at the DH. Training consisted of short, small group lectures (4-person audience) of 15 minutes or less, mixed with hands-on practical time with healthy volunteers from antenatal care. This option was employed to improve ability to get practice scans completed during a short period of time given the higher delivery volume at the DH compared with the HCs. After the 2-day intensive training, HC providers returned to their facility and were accompanied by a Master Trainer at least 2–3 days per week. Master Trainers assisted in mock enrollments and continued hands-on training until each learner reached their 25 proctored scans and could complete the OSCE. These Master Trainer visits to HCs were completed by week 3 post training. Shorter lectures, lack of clinical duties during training, increased practical hands-on experience, and an elongation of the hands-on training period over several weeks with use of Master Trainers were the main differences in the training schedule for the second cohort.

The ultrasound curriculum content for both the DH and HC trainings was identical (Table 2). It focused on identification of high-risk conditions in late pregnancy including fetal

**Table 2. Educational domains of the lectures and practical training.**

| Methods of Training | US Course | Training of Trainers Course |
|---|---|---|
| Core Lectures | • Ultrasound Knobs & Physics | • Principles of Adult Learning |
|  | • Finding Fetal Position | • Ultrasound Teaching Methods |
|  | • Fetal Heart Rate | • Hands-on Training Techniques |
|  | • Estimating Gestational Age: BPD, HC, FL, TCD | • Common Learner Mistakes |
|  | • Placenta Position | • Ultrasound Safety |
|  | • Fluid Volume |  |
|  | • Multiple Gestation |  |
| Small Group Didactics | Mini-lectures (<15 minutes, groups of 3–4) | • Live demonstration of common learner mistakes |
|  |  | • How to correct errors |
| Hands-on Practice: Live Models | Scan practice on volunteers from antenatal clinic | Master Trainer nurses were observed teaching new learners and feedback given |
| Mock Enrollments | 25 mock enrollments supervised by trainers | Master Trainer nurses supervised HC mock enrollments |
| Observed Structured Clinical Exam | Observed exams performed after 25 scans completed |  |

distress, non-vertex presentation, multiple gestation, abnormalities of the placenta and fluid, and methods of estimating gestational age. Guidance for diagnosis was provided on the study tool checklist which also served as a data collection tool for the parent study. For example, if fetal heart rate was less than 120bpm or greater than 160bpm by ultrasound, fetal distress was flagged. To assess oligohydramnios, the curriculum covered deepest vertical pocket (DVP) and amniotic fluid index (AFI). If DVP <2cm and AFI <8cm, trainees were taught to consider oligohydramnios versus ruptured membranes. Placenta previa was diagnosed if the inferior edge of the placenta approached or covered the internal cerival os. The entire uterus was scanned to assess for more than one fetus, while malpresentation was discerned if the head was not the presenting part. For gestational age, measurement of femur length (FL), biparietal diameter (BPD) and head circumference (HC) were collected to calculate estimated gestational age (EGA) using the software's automatic algorithm. Assessment of transcerebellar diameter was included solely as an exploratory measure.

Educational methods employed included home study using flash disk and printed materials including excerpts from the International Society of Ultrasound in Obstetrics and Gynecology (ISUOG) Manual [9], Partners In Health Manual of Ultrasound [10], video lectures and printed slides from lecture series. Training also included in-person short lectures, hands-on demonstrations using ultrasound equipment on healthy volunteers, hands-on live scanning practice on third trimester antenatal patients not in labor, and mock enrollments with active labor patients.

Several ultrasound trainers were engaged during both Phase 1 and Phase 2 trainings to ensure a 1:3 ratio during hands-on scanning activities. Trainers included a certified Ugandan sonographer, and several U.S.-based providers (obstetrics and gynecology resident physician, a family physician, and three emergency physicians with ultrasound/global health training). All U.S.-based ultrasound trainers had prior experience in POCUS in LMICs and availability during the training period. The Uganda sonographer was selected and supported by the in-country research team as part of the parent study. All trainers were debriefed on curricular content prior to training, and used the same guidelines, standard operating procedures, and checklist to implement hands-on training.

During and after the course for approximately 3 months, twice weekly communication was conducted between trainees, Master Trainers, and the study team using WhatsApp to trouble shoot, give personal feedback on scans, and identify potential problem areas for remediation. During the training period, the use of WhatsApp allowed for near real-time feedback to ensure accurate identification of high-risk conditions. The local sonographer trainer visited the sites weekly for 8 weeks to check machine maintenance and work on common scanning errors identified in remote image quality assurance (QA) review.

## Data collection

Trainees completed a survey to assess confidence level with clinical and ultrasound exams at three different timepoints: before training (pre), immediately after training (post) and 3 months after (follow-up). A 7-point Likert scale was used to determine confidence (1 = do not know how, 7 = extremely confident), and open response questions were presented to explore which learning resources were most valued by participants and prior use of technology in general.

During the two-week training, participants completed 25 proctored scans and subsequently performed an in-person Observed Structured Clinical Ultrasound Exam (OSCE, Appendix A). If the trainee received 80% correct, they passed the exam and were eligible to enroll patients for the parent study during regular work hours.

QA activities were completed for images generated by nurse and nurse midwife trainees; the two physicians who completed the training, surveys and OSCEs were not involved in enrolling participants for the parent study. Study images for the first 8 weeks after the initial training were stored and archived on the ultrasound machine then uploaded to a secure server cloud database. These images underwent blinded review by two independent expert sonographer clinicians (emergency ultrasound fellowship trained with extensive education experience, author SS and author KMM shared review duties). Images for Weeks 1–8 were reviewed for all four facilities. Images were scored individually for acceptability based on quality using the American College of Emergency Physicians (ACEP) Quality assurance 5-point grading scale [16] (1 and 2 are uninterpretable, while 3, 4, or 5 are adequate for interpretation), and any measurement errors were categorized individually as reasons to deem images unacceptable. After a relative plateau of acceptable image rate and error rate was achieved, a random selection of cases was chosen for determination of inter-rater reliability between the two QA reviewers.

Key informant interviews (KIIs) were conducted as part of a qualitative assessment of introducing POCUS in this context. Twelve KIIs (3 DH trainees, 3 DH Master Trainers, 6 HC trainees) with nurses and midwives from the four sites were conducted by a non-study team member in a confidential manner in a private location. Using open ended questions, the KII guide focused on: facilitators and barriers to ultrasound use, integration of ultrasound scan into routine work flow; health care workers' impressions of ultrasound training; challenges faced as part of the training; and women's acceptability of ultrasound scan. The guide was pilot tested prior to actual data collection with one DH trainee. Informed written consent was sought from participants. All interviews were audio-recorded and transcribed verbatim. Notes taken to during interviews were used to supplement the recordings. Each interview lasted between 45 to 60 minutes. The two physicians were not included in the qualitative component given that the nurse midwife trainees led daily ultrasound use.

## Data analysis

Descriptive statistics were used to portray respondents' answers to the Likert-scale survey questions, OSCE data and blinded QA review measures. Two expert reviewers reviewed the images submitted by the participant clinicians to determine agreement of whether the images were either "acceptable" (ACEP score of 3–5), or "unacceptable" (ACEP score of 1,2 or with an identifiable error in measurement) in the final week of the study period after 8 weeks of training and presumed plateau of sonography skills. Cohen's kappa for agreement was calculated to assess inter-rater reliability between the two expert reviewers.

The transcribed KII scripts were entered into a Microsoft Word processing program in preparation for data analysis and scripts reviewed by multiple readers who searched them for predetermined themes and classified them into categories. Data analysis was done using the content analysis method. An excel analysis matrix table was developed and structured under each study objective. Data coding was done into the matrix table for each thematic area under each objective to identify information or issues coming up under each theme. The transcripts were read and re-read to identify the emerging themes. All data relevant to each category was identified and examined using the process of constant comparison, in which each item was checked or compared with the rest of the data in order to establish analytical categories. Rich and relevant textual quotes were identified and used to support the emerging themes and categories of data respectively.

## Ethical considerations

All training participants and key informants provided informed written consent. This study was IRB approved by the University of California San Francisco IRB (17–22310), and the

Higher Degrees, Research and Ethics Committee at Makerere University in Uganda (protocol # 515).

## Results

All 25 participants had never used ultrasound or received formal ultrasound training before. All nurse midwifes used mobile phones regularly (daily), however had less access to laptop or desktop computers. Technology use in general was relatively new for all participants; none of the nurses had used the internet for >5 years, and 2 nurses first ever use of the internet was within 1 year of the training.

Of the 23 nurse midwives who completed the training, 21 began enrolling patients after passing their OSCE and performed 558 total studies over the 8-week evaluation period. The 2 physicians trained did not enroll patients for the parent study.

### Changes in confidence: Survey results

Of the 25 total trainees, we received a total of 17 completed pre-course surveys [9 DH, 8 HC], 17 post-course surveys [9 DH and 8 HC] and 18 follow up surveys [7 DH, 11 HC]. We were unable to collect all surveys due to competing clinical duties and administrative challenges.

Changes in confidence over time are demonstrated in Fig 1. Pre-course, none of the trainees had ever performed any ultrasound exams, and all rated themselves as "not confident, 1" on the Likert scale assessing comfort with ultrasound exams for use of knobs, gain and depth for image optimization, or making any measures as part of the late pregnancy ultrasound protocol for high-risk conditions. Immediately post-course, confidence of the participants improved (from 1) for measuring fetal heart rate (6.63, stdev 0.5, n = 16), assessing malpresentation (6.63, stdev 0.5, n = 16), identifying multiple gestation (6.06, stdev 0.77, n = 16), placenta previa (5.60, stdev 0.99, n = 15), and oligohydramnios (6.06, stdev 0.77, n = 16). Measures of gestational age also showed increase in confidence levels (from 1 pre-course for all measures) to 6.07 [stdev 0.80, n = 15] for BPD, 6.13 for HC [stdev 0.81, n = 16], and 3.80 [stdev 1.41,

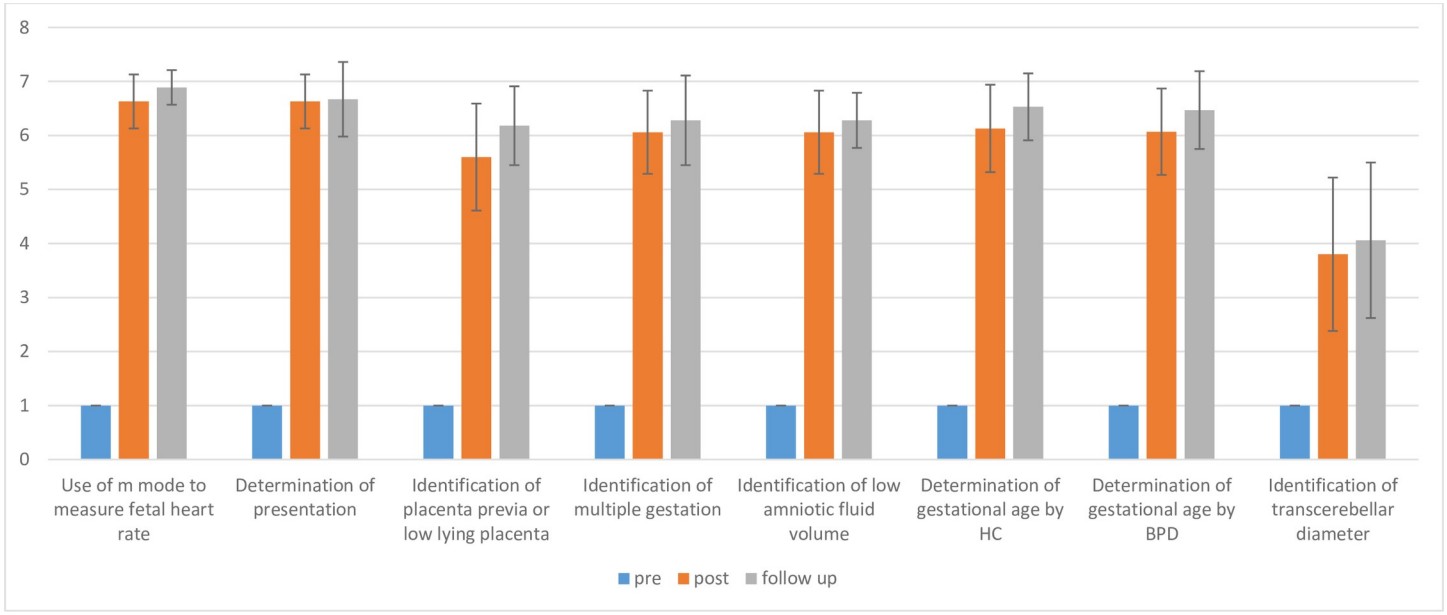

**Fig 1. Changes in learner confidence pre-course, immediately post-course, and on 3-month follow-up.**

n = 15] for TCD. Changes in confidence were similar among Phase 1 and Phase 2 trainees. Improvements in confidence were sustained and all measures improved further on 3-month follow up survey.

At 3-month follow up, clinicians reported completing an average of 80 scans since their initial training (range 30–250) with several reporting simply >100 (n = 4) which were not included in the average.

Clinicians stated that they used resources for enhancing their post-course learning, including the WhatsApp group (12/16), asking colleagues (e.g. other trainees, Master Trainers or the local consultant) (12/16), excerpts from the textbooks (10/16), a mobile app (3/16) and websites (2/16). No clinicians indicated they used podcasts or other resources.

## Training efficacy: OSCE

OSCE results are summarized in Table 3. Of the 25 who took their OSCE exams, 10/11 from the DH and 12/14 from the HC passed (>80%) on their first attempt. After several (2–5 per trainee) focused one-on-one scan sessions with the local sonographer trainer (approximately 8 hours), 1 DH trainee was still unable to pass the OSCE and stopped enrolling patients, but all remaining nurses from the HC were able to pass the OSCE and able to enroll patients. Among those who passed on their first attempt, the average passing initial OSCE score was 90.4% [85.3–94.1%] at the HCs (n = 12 passing, n = 2 fails with scores of 65% and 79%), and 88.2% [81.8–93.9%] at the DH (n = 10 passing, n = 1 fail 66.7%).

Of the high-risk conditions, first-attempt OSCEs revealed 92% (23/25) could correctly measure fetal heart rate, 100% (25/25) were able to identify fetal position, 92% (23/25) could correctly follow the placental edge to the most inferior border to assess for previa, and 84% (21/25) were able to identify and measure the DVP of amniotic fluid. EGA measures proved more difficult: correct measures for HC 52% (13/25), BPD 56% (14/25), FL 48% (12/25), and TCD 12% (3/25) were performed during initial post course OSCE exams. Correct use of the aggregate report for EGA was performed 92% (23/25) of the time.

**Table 3. Observed structured clinical ultrasound exam results.**

| OSCE Results | District Hospital (n = 11) | Health Centers (n = 14) |
|---|---|---|
| Participants Passing Exam | 10 | 14 |
| Average Final Passing Score (First Attempt) | 88.2% | 90.4% |
| **OSCE Component** | **Total Number Performing Correctly (n = 25)** | **Total Percent Performing Correctly** |
| Fetal Heart Rate | 23 | 92% |
| Fetal Position/ Presentation | 25 | 100% |
| Placenta Location | 23 | 92% |
| Deepest Vertical Pocket of Amniotic Fluid | 21 | 84% |
| EGA: HC | 13 | 52% |
| EGA: BPD | 14 | 56% |
| EGA: FL | 12 | 48% |
| EGA: TCD | 3 | 12% |
| EGA: Use of Aggregate Report function | 23 | 92% |

## Training efficacy: Image review

Acceptability of images for both overall quality of image and measurements on the image was calculated for each portion of the ultrasound protocol and are reported in Table 4. For each part of the POCUS protocol, quality and measurement ability over time improved among both training cohorts. Overall kappa between two blinded reviewers was 1.0 for fetal heart rate, head position, fluid and FL, and between 0.829–0.928 for all other measures. Commonly made errors and rate of these errors occurring over time are shown in Fig 2, including uninterpretable image quality, wrong plane of measurement (e.g., coronal plane vs correct axial plane of measurement for BPD/HC), and placement of calipers over-estimating gestational age (e.g. incorrect, too wide fit of circular calipers around head). We also measured rates of underestimation of gestational age (e.g. too narrow fit of calipers), and wrong structure measured in the correct plane (e.g. bladder of fetus instead of amniotic fluid); however, these represented less than 0.2% of all images reviewed and were not included in the analysis.

## Perceptions of POCUS training

Several themes emerged from the open-ended survey questions, including learner perceptions of the most successful training techniques and recommendations for improved training in the future. Through KII, key themes regarding valued components of the training and learner considerations for course improvement were identified. Overall, nurses felt they wished they had more ability to practice ultrasound skills aside from immediately during patient care including time available for learning and availability of ultrasound machine (time, location of ultrasound machine).

Trainees at both the DH and HC felt having written materials was useful.

*"We have a training manual which has everything that we need to know about ultra sound. Whenever we could get a challenge, we could always refer to that book." (HC respondent)*

*"They gave us an ultrasound scan manual for the whole package and even we were given flashes." (DH respondent)*

Trainees from both the HC and DH expressed how valuable hands-on practice.

*"About hands on practice, we were shown how to hold the probe, how to position the mother when you are taking the DVP [deepest vertical pocket] and also the steps to follow while you are doing the scan." (DH respondent) "*

**Table 4. Accuracy of images over time.**

| Us Protocol Component | Weeks 1–4 | | | Weeks 5–8 | | | (Week 8) Kappa | |
| --- | --- | --- | --- | --- | --- | --- | --- | --- |
| | # Images Reviewed | Quality | Measure Acceptable | # Images Reviewed | Quality | Measure Acceptable | Quality | Measure Acceptable |
| FHR | 262 | 100% | 100% | 224 | 100% | 100% | 1 | 1 |
| EGA: BPD | 255 | 80% | 80% | 204 | 89% | 89% | 0.83 | 0.829 |
| EGA: HC | 253 | 82% | 66% | 204 | 88% | 79% | 0.848 | 0.92 |
| EGA: FL | 242 | 97% | 95% | 200 | 99% | 99% | 1 | 1 |
| EGA: TCD | 4 | 75% | 75% | 8 | 80% | 80% | *too few to calculate | |
| Head Position | 259 | 99% | n/a | 218 | 98% | n/a | 1 | n/a |
| Placenta Location | 246 | 98% | | 195 | 98% | | 1 | n/a |
| Amniotic Fluid | 233 | 97% | | 200 | 99% | | 0.928 | n/a |

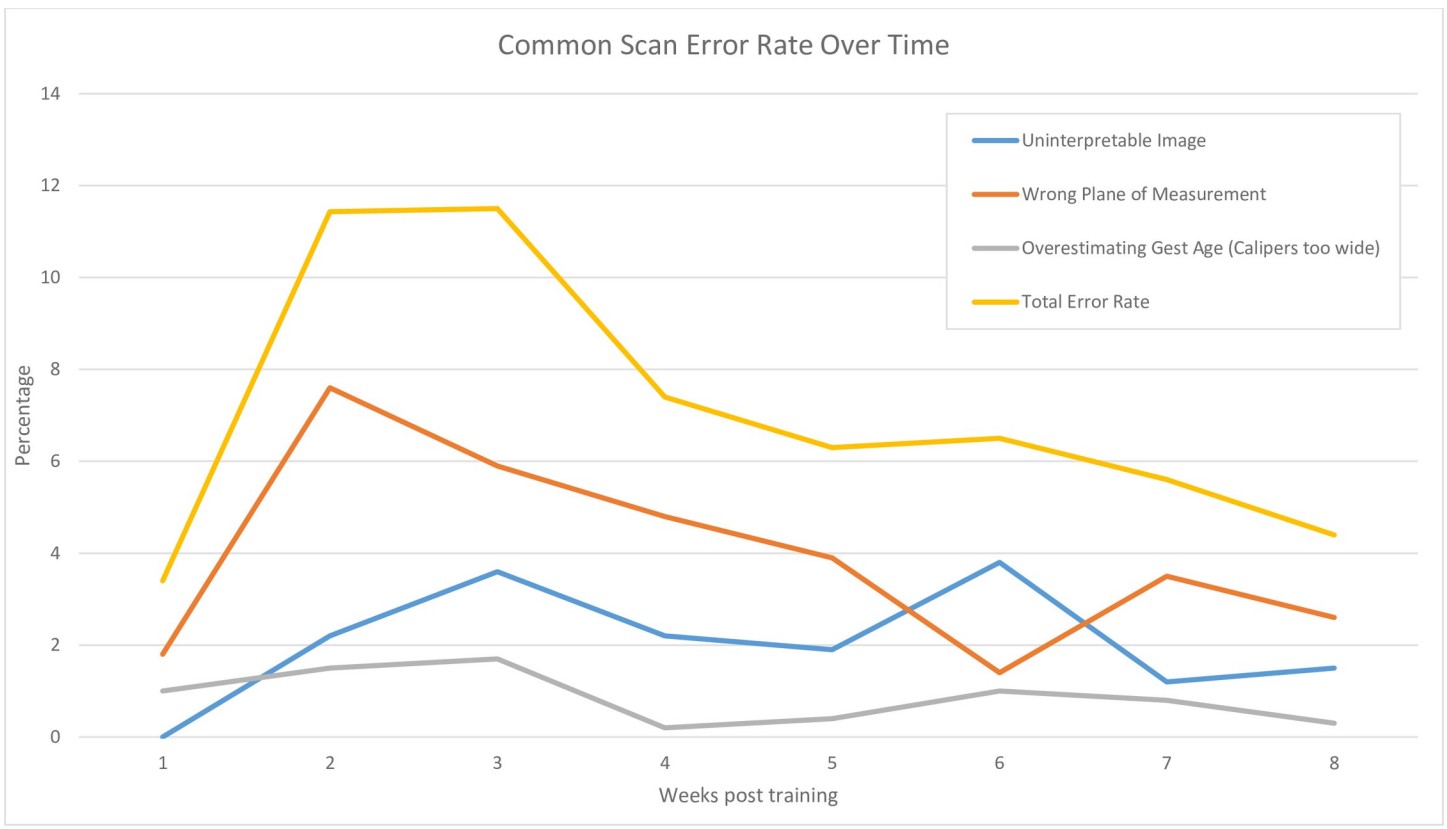

**Fig 2. Common errors in ultrasound performance and error rate over time after training.**

*"During the theory part, they introduced the scan informing us the parameters we shall be measuring that is FA, HC, BPD theoretically. When it came to practical part, they showed us what we had talked about in theory." (HC respondent)*

HC trainees noted how mentoring visits, in particular, enhanced their skills.

*"The mentors would hold your hand and show you how to scan the mother." (HC respondent)*

*"For instance during those visits when the mentor came, I was not sure of locating the placenta and it would confuse me, but when I got used with that mentor, this was benefit to me, even looking for the femur length FL) of the baby, measuring the HC (head circumference), BPD (biparietal diameter)" (HC respondent)*

Three Master Trainer nurses were interviewed to determine the impact of being chosen for that role on their clinical and teaching skills, as well as self-esteem and professional attitude. These three Master Trainers unanimously agreed that their role as a trainer for their nurse midwife colleagues at the HCs was a positive experience, allowing improved skills as an educator as well as positive influence on self-esteem.

*"I have attained more knowledge and skills from the TOT role, even it has helped us to improve on the way we plan the management of these mothers." (Master Trainer Respondent 1)*

*"In relation to skills, I have admired the skills attained in ultrasound scan to the extent that I am thinking of changing my career to a radiographer instead of nurse. I feel I am capable."* *(Master Trainer Respondent 2)*

*"I have got prior knowledge and have even become known in other facilities which I had never gone to before such as [X] HC III because I am a TOT, so I have been recognized by other health workers because of the scan study" (Master Trainer Respondent 2)*

Strengths of the training were identified as friendly trainers, novel information/subject matter, and award of certificates at the end of the training.

*"The facilitators were friendly and could listen to us whether you are slow learner or what, they could consider everyone and give us time to learn. This helped us to continue." (HC respondent)*

Challenges identified in the training were short duration of the course, limited time for hands-on training, competing priorities during the training with clinical duties and learning duties.

*"Limited training time because when you look at the sonographers who are trained for the scan, they take a lot of time to study and learn how to examine mothers but now for us, we were only trained for three days." (HC respondent)*

A few participants felt it was difficult to understand the American accents of the English-speaking trainers, and easier to understand the Ugandan sonographer trainer (also English speaking).

Of note, all 12 key informants wished gender identification could have been included in the training, to be able to answer the most common questions mothers ask regarding their ultrasound scans. Topics identified for requested further training included additional training for cerebellar measurement for EGA, placenta abnormalities including abruption and grading, early pregnancy measures for EGA and ectopic pregnancy, gender identification, and fetal weight.

## Discussion

Our study demonstrates that nurse midwives in a LMIC context were able to build confidence and skills after an ultrasound training course and accurately perform ultrasound during labor triage to detect high-risk conditions in pregnancy. We also gathered valuable qualitative data to inform the ultrasound education community in best practices.

Despite our novice cohort of learners, the nurse midwives in our study had a substantial improvement in confidence for all aspects of the ultrasound protocol, which was sustained even at the 3-month follow up. Immediately after the two-week training course, we had a high rate of participants able to pass the OSCE, demonstrating attained skills. Image review showed that accuracy improved over time, with decreased overall rate of images with errors to under 8% for the last month of the study period, and under 5% for the eighth week of review. There was a high level of agreement (Cohen's kappa 0.8–1.0 for all parts of the scan protocol) among expert reviewers on the acceptability of the images and measures performed by our study midwives.

Regarding the skills attained using our training protocol, our findings differ from the study by Kimberly et al [14], whereby nurse midwives demonstrated a much higher error rate for

EGA measures of the head (up to 70%) and FL was not well mastered in that training cohort. For that study, most scans were conducted in the second or third trimester and training comprised three 2-week training periods interspersed by 2–3 months of minimally supervised independent scanning. In our study, nurse midwives readily adopted ability to perform fetal heart rate and identify fetal distress, find the fetal head and discern fetal position, search for multiple gestations, find placental location and assess for previa, and assess amniotic fluid for oligohydramnios. Measurement of GA-related biometry was also well-mastered by the end of the training, though common mistakes identified included measuring BPD and HC in a non-axial plane. FL was well performed by our nurse midwives and given that accuracy of this measure is not affected by coning of the fetal head during labor, FL may warrant further study as an easier measurement during labor. The cerebellum for transcerebellar diameter proved difficult to visualize and measure in active labor, perhaps due to fetal head compression, and shadowing due to the fetal skull or the mother's pubic bone. This was an exploratory measurement for this study; while our data show that this may not be a feasible measurement at triage, it may be interesting to continue to explore TCD as a measure for gestational age in non-labor contexts [17]. Our training approach which adopted a 2-week initial training period with robust hands-on mentorship followed by frequent QA interactions through both WhatsApp and in-person mentorship may have contributed to enhanced uptake of skills.

Through interviews with our two training cohorts, we identified a number of factors to promote as best practice. Despite different training procedures at the DH and HCs, themes were similar. First, the use of local trainers for language purposes and contextual understanding was important. Our model of integrating a TOT model improved hands-on practice for HC trainees, promoted confidence among the 3 DH Master Trainers and improved communication between the HC and DH midwives. The parent study compensated these Master Trainers for their time and transportation to the HCs, which warrants consideration for future TOT models. Second, a trainer to trainee ratio of 1 to 4 or better is recommended to allow ample hands-on time for practical skills. Third, shorter lectures (approximately 15 min) in smaller groups were easier to administer without the need for projector equipment or electricity given laptop battery life, and learners were generally more engaged in smaller groups. Fourth, although nurses requested longer duration of training, our results demonstrate that both skill and confidence improved to an acceptable level in our two-week training with quality assessment and mentorship was successful. And finally, given internet use was relatively sparse among the nurses, the written materials are a valuable resource.

We faced several unanticipated challenges in our training evaluation. There were delays in sending image feedback to nurses due to inability to upload images to a server without internet access and power outages which affected TOT/local sonographer visits. Additionally, the cost associated with use of WhatsApp for nurses to receive messages and feedback on images was compensated by the parent study, which may have implications for generalizability and sustainability as a routine method for distance POCUS education. Key informants also noted that they felt torn between nursing clinical duties and ability to receive ultrasound training, especially during times of high patient volume or lower clinician turn-out.

## Conclusions

In summary, these data suggest that a two-week intensive training course with short lectures and long periods of hands-on training, followed by ongoing mentoring by local trainers improves confidence and obstetric POCUS skills for identification of critical conditions in labor. Nurse midwives in rural Uganda can accurately perform ultrasound at labor triage to detect fetal distress, and placenta previa, in addition to high-risk conditions such as multiple

gestation, low amniotic fluid, and can estimate gestational age using a variety of fetal biometry measures.

## Supporting information

**S1 File. OSCE appendix.**
(DOCX)

## Acknowledgments

This study was part of the East Africa Preterm Birth Initiative (PTBi-EA), a multi-year, multi-country effort. We would like to acknowledge the time and effort put forth by our trainers, Drs. Jorge Garcia, Scott Owens and Nicole Teal. We thank Nathan Isabirye and Innocent Inhensiko for administrative coordination, data collection and management, as well as Dr. Peter Waiswa for his advisory support and leadership. We deeply appreciate the volunteer patients from antenatal clinic who allowed our learners to practice ultrasound during their visits.

## Author Contributions

**Conceptualization:** Sachita Shah, Nicole Santos, Jude Mulowooza, Dilys Walker.

**Data curation:** Sachita Shah, Nicole Santos, Rose Kisa, Krithika Meera Muruganandan.

**Formal analysis:** Sachita Shah, Nicole Santos, Rose Kisa, Krithika Meera Muruganandan.

**Funding acquisition:** Dilys Walker.

**Investigation:** Sachita Shah, Nicole Santos, Odida Mike Maxwell, Jude Mulowooza, Dilys Walker, Krithika Meera Muruganandan.

**Methodology:** Odida Mike Maxwell, Jude Mulowooza, Dilys Walker, Krithika Meera Muruganandan.

**Project administration:** Sachita Shah, Nicole Santos, Odida Mike Maxwell, Jude Mulowooza, Dilys Walker, Krithika Meera Muruganandan.

**Resources:** Sachita Shah, Nicole Santos.

**Software:** Nicole Santos.

**Supervision:** Sachita Shah, Jude Mulowooza, Dilys Walker, Krithika Meera Muruganandan.

**Writing – original draft:** Sachita Shah, Nicole Santos, Rose Kisa.

**Writing – review & editing:** Nicole Santos, Rose Kisa, Odida Mike Maxwell, Jude Mulowooza, Dilys Walker, Krithika Meera Muruganandan.

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
