## [Decision Letter · Decision Letter 0]

21 Apr 2020

PONE-D-20-03332

Efficacy of an ultrasound training program for nurse midwives to assess high-risk conditions at labor triage in rural Uganda

PLOS ONE

Dear Dr. Shah,

Thank you for submitting your manuscript to PLOS ONE. After careful consideration, we feel that it has merit but does not fully meet PLOS ONE’s publication criteria as it currently stands. Therefore, we invite you to submit a revised version of the manuscript that addresses the points raised during the review process.

We would appreciate receiving your revised manuscript by Jun 05 2020 11:59PM. To enhance the reproducibility of your results, we recommend that if applicable you deposit your laboratory protocols in protocols.io, where a protocol can be assigned its own identifier (DOI) such that it can be cited independently in the future. For instructions see: http://journals.plos.org/plosone/s/submission-guidelines#loc-laboratory-protocols

We look forward to receiving your revised manuscript.

Kind regards,

Sara Ornaghi, M.D., Ph.D.

Academic Editor

PLOS ONE

2. Please provide additional details regarding participant consent. In the ethics statement in the Methods and online submission information, please ensure that you have specified whether consent was informed.

3. PLOS ONE will consider submissions that present new methods, software, or databases as the primary focus of the manuscript if they meet the criteria of utility, validation, and availability described here: http://journals.plos.org/plosone/s/submission-guidelines#loc-methods-software-databases-and-tools. To meet these criteria, please provide supporting materials enabling other teachers and researchers to replicate your teaching intervention such as sample worksheets, a detailed lesson plan or curriculum or other educational materials. If you include supporting materials, they should not be under a copyright more restrictive than CC-BY.

Reviewers' comments:

Reviewer's Responses to Questions

**Comments to the Author**

1. Is the manuscript technically sound, and do the data support the conclusions?

Reviewer #1: Yes

Reviewer #2: Partly

2. Has the statistical analysis been performed appropriately and rigorously? 

Reviewer #1: Yes

Reviewer #2: I Don't Know

3. Have the authors made all data underlying the findings in their manuscript fully available?

Reviewer #1: Yes

Reviewer #2: Yes

4. Is the manuscript presented in an intelligible fashion and written in standard English?

Reviewer #1: Yes

Reviewer #2: Yes

5. Review Comments to the Author

Reviewer #1: The aim of the study was to demonstrate that nurse-midwives were able to build confidence and skills after an ultrasound training course to detect high risk conditions during labor triage. This intervention is particularly effective in the study setting (low income country) to improve maternal and fetal outcomes during labour, with a relevant impact on public health.

Therefor the qualitative analysis allowed to identify best practices for training.

However there are some aspects that should be explored or clarified:

1. In the quantitative analysis is unclear how the sample size was defined. I suggest to better describe the inclusion and exclusion criteria of healthcare professional included. It should be defined the selection process of subjects included into research, and their characteristics in term of professional experience, attitude,.... This aspect is needed to extend the results in other setting.

2. In the qualitative analysis it should be describe the result in both groups (district hospital and health centers) because they received a different training course protocol.

I suggest to explore deeply the experience of "training of trainers"

Reviewer #2: This is a mixed-method study to evaluate a 2 weeks ultrasound training protocol for nurses, midwives and doctors in rural Uganda in order to detect seven high-risk conditions in labor: fetal demise, fetal distress, malpresentation, multiple gestation, placenta previa, oligohydramnios and preterm delivery.

This a very important topic because many high-risk conditions of pregnancy are undetected until the time of delivery in low-income countries

This study demonstrate that nurse midwives without previous ultrasound experience can detect high-risk conditions during labor with a high rate of quality and accuracy after training

Here some comments to the authors.

Materials and Methods

- Line 154: the authors report that in the public district hospital there were 20% of deliveries complicated; what were the complications observed?

- Line 160: “complicated patients from these sites are referred to the district hospital”. What kind of complicated patients are referred to the district hospital and which patients are managed in the regional health centers?

- Line 177: What are the specific differences between phase 1 and phase 2? It may be helpful to enter a table.

- Line 184-187: please specify the criteria and guidelines used to diagnose fetal distress and other indicated risk conditions. It may be helpful to enter a table.

- Tab 1: I'd change the table. Since the differences and characteristics of the training of phase 1 and phase 2 have already been reported previously, at this point I would insert in the table only the data related to the training of trainers course. About the training course for trainers...did only the three midwife nurses mentioned above who had participated in phase 1 or did someone else participate in this course?

- Line 200: How were the ultrasound trainers chosen? Did all ultrasound trainers use the same guidelines and reference methods?

- Line 205: “TOT master trainers” please specify who they are and how they were chosen

- Line 226-227: Why were these two different time intervals chosen? to make sure that there were more or less the same number of births between the district hospital and the HCs? And how many births have there been at these intervals of time?

- Line 229-230: Why weren't the two doctors involved?

- Line 248: Why weren't all the participants interviewed?

Results

- Line 291-300: The characteristics of the participants should be put into materials and methods (“ participants ” line 165-171)

- Line 311-332: The characteristics and differences between phase 1 and phase 2 should be put into materials and methods

- Line 336-338: Specify why there is no data on all trainees

- Line 360: “ asking other people”: specify who these people are

- Table 3: why the average final passing score for health centers is 90.4%?

- With respect to the reported results, it might also be useful to make an analysis of the differences between phase 1 and phase 2.

Discussion

- Line 495: How many and what complications have the trainees diagnosed? Did they miss any complications? This data should also be put into materials and methods

- Line 510-519: I would describe the differences from the study mentioned: in the study mentioned how was the training organized? Was the ultrasound performed in labor or in which trimester of pregnancy?

- Line 516: What are the parameters used to establish the gestational period? Which curves are used?

6. PLOS authors have the option to publish the peer review history of their article (what does this mean?). If published, this will include your full peer review and any attached files.

Reviewer #1: No

Reviewer #2: No

---

## [Author Response · Author response to Decision Letter 0]

15 May 2020

Reviewer #1

The aim of the study was to demonstrate that nurse-midwives were able to build confidence and skills after an ultrasound training course to detect high risk conditions during labor triage. This intervention is particularly effective in the study setting (low income country) to improve maternal and fetal outcomes during labour, with a relevant impact on public health. Therefor the qualitative analysis allowed to identify best practices for training. However there are some aspects that should be explored or clarified:

1. In the quantitative analysis is unclear how the sample size was defined. I suggest to better describe the inclusion and exclusion criteria of healthcare professional included. It should be defined the selection process of subjects included into research, and their characteristics in term of professional experience, attitude. This aspect is needed to extend the results in other setting.

Thank you for this recommendation. The number of trainees was determined by the larger study within which this training program was nested. The trained nurse midwives led all enrollment and clinical assessments of study participants. We trained enough nurse midwives to ensure that each shift was covered by at least one study-trained midwife and one research nurse. We have clarified the size of the training cohort in the participant section of the methods section (lines 159-163). 

2. In the qualitative analysis it should be describe the result in both groups (district hospital and health centers) because they received a different training course protocol. 

We appreciate this comment. We have included quotes from both the district hospital (DH) and health center (HC) trainees in the qualitative section of the results. We have noted in the results section where themes overlapped between the two groups despite the different training delivery models versus where HC-specific themes arose. 

3. I suggest to explore deeply the experience of "training of trainers."

We appreciate this suggestion and have added some additional exploration of the TOT model. Specifically, in the results, we included qualitative quotes from the Master Trainers that shed light on how their role enhanced their confidence and skills (beginning line 471). In the discussion (lines 560-564), we explore benefits of this type of training model, but also key considerations for replicability. Lastly, we also provide more information regarding TOT selection in the methods, as well as how they were essential to Phase 2 training (lines 186-190).

Reviewer #2

This is a mixed-method study to evaluate a 2 weeks ultrasound training protocol for nurses, midwives and doctors in rural Uganda in order to detect seven high-risk conditions in labor: fetal demise, fetal distress, malpresentation, multiple gestation, placenta previa, oligohydramnios and preterm delivery. This a very important topic because many high-risk conditions of pregnancy are undetected until the time of delivery in low-income countries. This study demonstrate that nurse midwives without previous ultrasound experience can detect high-risk conditions during labor with a high rate of quality and accuracy after training.

Here some comments to the authors.

Materials and Methods

4. Line 154: the authors report that in the public district hospital there were 20% of deliveries complicated; what were the complications observed?

Thank you for this question. We have clarified the complications of interest of the parent study this training was a part of (lines 132-134). The conditions include preterm birth, oligohydramnios, placenta previa, multiple gestation, malpresentation, fetal distress or stillbirth.

5. Line 160: “complicated patients from these sites are referred to the district hospital”. What kind of complicated patients are referred to the district hospital and which patients are managed in the regional health centers?

We have clarified in the text that the six conditions of interest (per the parent study) warrant referral from the health center to the district hospital, unless delivery is imminent (lines 148-150). It should be noted that other conditions that were not assessed as part of this study, such as antepartum hemorrhage, eclampsia and prior Caesarean, warrant referral. 

6. Line 177: What are the specific differences between phase 1 and phase 2? It may be helpful to enter a table.

Thank you for this question. The curriculum itself did not differ between the two trainings; both focused on detection of the conditions outlined in Table 1. However, training implementation varied between Phase 1 and Phase 2. We have revised this for clarity and moved our description of how the implementation differed from the results section to the methods section (per comment #14 below).

7. Line 184-187: please specify the criteria and guidelines used to diagnose fetal distress and other indicated risk conditions. It may be helpful to enter a table.

Per the reviewer's recommendation, we have included the criteria for how ultrasound was used to diagnosis the conditions of interest (lines 220-233). We have also clarified how these guidelines were included on a clinical assessment/ultrasound checklist that was part of the parent study.

8. Tab 1: I'd change the table. Since the differences and characteristics of the training of phase 1 and phase 2 have already been reported previously, at this point I would insert in the table only the data related to the training of trainers course. About the training course for trainers...did only the three midwife nurses mentioned above who had participated in phase 1 or did someone else participate in this course?

Thank you for the suggestion. Since the curriculum is the same between Phase 1 and Phase 2, we retained Table 1 and have instead clarified the preceding text that the implementation process differed (see comment #6). 

Regarding the second question – three midwife nurses from Phase 1 were trained as ToT master trainers. This is because they would later serve as trainers to the HC-trainees in Phase 2 whereby one Trainer would be assigned to one HC. We have included this in the text for additional context (lines 186-190).

9. Line 200: How were the ultrasound trainers chosen? Did all ultrasound trainers use the same guidelines and reference methods?

We have added additional information regarding selection of trainers (lines 251-256). U.S.-based ultrasound trainers were selected by the lead ultrasound consultant based on experience in point of care ultrasound in low-resource settings, availability and interest. The Ugandan sonographer was hired by the larger study. All trainers utilized the same guidelines, procedures and study checklist.

10. Line 205: “TOT master trainers” please specify who they are and how they were chosen

We have provided additional information about how these Master Trainers were selected (lines 186-190). Additionally, please see Comment #3 above where we have addressed another's reviewer's desire to hear more about this component of the training program. 

11. Line 226-227: Why were these two different time intervals chosen? to make sure that there were more or less the same number of births between the district hospital and the HCs? And how many births have there been at these intervals of time?

The two different training intervals were chosen based on sample size attainment of the parent study, clarified in lines 182-184. Since the DH had higher delivery volumes, they reached sample size faster thus, leading to ultrasound training in October 2018. The HCs had smaller delivery volumes, so ultrasound could not yet be introduced until January 2019. We have also provided some additional information about the parent study in which this training was rolled out to provide more context (lines 160-164).

12. Line 229-230: Why weren't the two doctors involved?

We appreciate this point. The parent study felt that the two doctors should be trained in ultrasound so that they were familiar with the parent study protocols given the hierarchies in the maternity ward. However, neither doctor was involved in the parent study enrollment and clinical assessment of participants. The interventions being tested were designed to be administered by midwives. This is why the doctors were not involved in image QA activities or the key informant interviews. We have noted this in lines 281-282 and lines 307-309.

13. Line 248: Why weren't all the participants interviewed?

The parent study aimed to obtain 12 interviews from the outset (3 DH midwives; 3 DH TOT trainers; 6 HCs trainees) for the qualitative component as we believed this would be sufficient for thematic saturation. The 12 were chosen based on availability; those excluded were unavailable due to scheduling, staff transfers to other facilities and/or maternity leave. We also chose to exclude the two physicians since they did not perform daily ultrasound activities (lines 307-309), as well as one nurse midwife trainee with whom the interview guides were pilot tested (line 304). 

Results

14. Line 291-300: The characteristics of the participants should be put into materials and methods (“ participants ” line 165-171)

Thank you for this guidance. We have moved information regarding the trainee characteristics into the Methods section as recommended (lines 165-176). We also noticed that Table 1 was incorrect (n=23) and have corrected the n for provider type based on the corresponding text (n=25). Please note that because the data regarding prior technology use was garnered from survey data, we have retained it in the results section. 

15. Line 311-332: The characteristics and differences between phase 1 and phase 2 should be put into materials and methods

Thank you for this suggestion. We have moved this information to the methods as suggested by the reviewer and believe it greatly improves the clarity and flow (lines 192-215).

16. Line 336-338: Specify why there is no data on all trainees

We have integrated into the text an explanation (lines 354-355) for missing data citing competing clinical priorities and administrative challenges, e.g. inability to print surveys, staff transfer at follow-up timepoints. 

17. Line 360: “ asking other people”: specify who these people are

We have clarified in the text that "other people" constitute other trainees, Master Trainers and the local Uganda consultant when available (lines 380-381). 

18. Table 3: why the average final passing score for health centers is 90.4%?

We have clarified in the text (lines 391-392) and Table 3 that the passing score is presented among those who passed on their first attempt. 

19. With respect to the reported results, it might also be useful to make an analysis of the differences between phase 1 and phase 2.

Thank you for this comment. We found that changes in confidence were similar across the two groups in preliminary analysis and have noted this in the text (lines 368-369). For image review/QA data, given the small number of trainees and that the curriculum itself was identical, we opted to present results among the entire cohort. OSCE data and KII data were previously presented by DH versus HC cohorts to highlight differences between the phases. 

Discussion 

1. Line 495: How many and what complications have the trainees diagnosed? Did they miss any complications? This data should also be put into materials and methods

The complications of interest are now further clarified in the methods section (lines 132-134). We have also described how the WhatsApp groups were used to aid in complication diagnosis during the training period (lines 261-263). Please note that diagnostic accuracy of these conditions (after the training period) is examined in two forthcoming manuscripts associated with the parent study outcomes. The DH study manuscript is currently under review; the HC study manuscript will be submitted by the end of this month.

2. Line 510-519: I would describe the differences from the study mentioned: in the study mentioned how was the training organized? Was the ultrasound performed in labor or in which trimester of pregnancy

We appreciate this recommendation and have further elaborated on how the Kimberly et al training differed from the training program presented. The former is summarized in lines 536-538, while the latter is described in lines 552-555. 

3. Line 516: What are the parameters used to establish the gestational period? Which curves are used?

We have clarified in the methods (lines 230-233) which fetal biometry measures were used to assess estimated gestational age. In the discussion, we have edited the verbiage to improve clarity.

---

## [Decision Letter · Decision Letter 1]

12 Jun 2020

Efficacy of an ultrasound training program for nurse midwives to assess high-risk conditions at labor triage in rural Uganda

PONE-D-20-03332R1

Dear Dr. Shah,

We’re pleased to inform you that your manuscript has been judged scientifically suitable for publication and will be formally accepted for publication once it meets all outstanding technical requirements.

Kind regards,

Sara Ornaghi, M.D., Ph.D.

Academic Editor

PLOS ONE

Additional Editor Comments (optional):

Reviewers' comments:

Reviewer's Responses to Questions

**Comments to the Author**

1. If the authors have adequately addressed your comments raised in a previous round of review and you feel that this manuscript is now acceptable for publication, you may indicate that here to bypass the “Comments to the Author” section, enter your conflict of interest statement in the “Confidential to Editor” section, and submit your "Accept" recommendation.

Reviewer #2: All comments have been addressed

2. Is the manuscript technically sound, and do the data support the conclusions?

Reviewer #2: (No Response)

3. Has the statistical analysis been performed appropriately and rigorously? 

Reviewer #2: (No Response)

4. Have the authors made all data underlying the findings in their manuscript fully available?

Reviewer #2: (No Response)

5. Is the manuscript presented in an intelligible fashion and written in standard English?

Reviewer #2: (No Response)

6. Review Comments to the Author

Reviewer #2: (No Response)

7. PLOS authors have the option to publish the peer review history of their article (what does this mean?). If published, this will include your full peer review and any attached files.

Reviewer #2: No

---

## [Editor Report · Acceptance letter]

19 Jun 2020

PONE-D-20-03332R1 

Efficacy of an ultrasound training program for nurse midwives to assess high-risk conditions at labor triage in rural Uganda 

Dear Dr. Shah:

I'm pleased to inform you that your manuscript has been deemed suitable for publication in PLOS ONE. Congratulations! Your manuscript is now with our production department. 

Kind regards, 

on behalf of

Dr. Sara Ornaghi 

Academic Editor

PLOS ONE